# C-type natriuretic peptide improves maternally aged oocytes quality by inhibiting excessive PINK1/Parkin-mediated mitophagy

**Hui Zhang[1,2†], Chan Li[1,2†], Qingyang Liu[1,2], Jingmei Li[1,2], Hao Wu[1,2], Rui Xu[1,2], Yidan Sun[1,2], Ming Cheng[1,2], Xiaoe Zhao[1,2], Menghao Pan[1,2], Qiang Wei[1,2]\*, Baohua Ma[1,2]\***

[1]College of Veterinary Medicine, Northwest A&F University, Yangling, China; [2]Key Laboratory of Animal Biotechnology, Ministry of Agriculture, Yangling, China

**Abstract** The overall oocyte quality declines with aging, and this effect is strongly associated with a higher reactive oxygen species (ROS) level and the resultant oxidative damage. C-type natriuretic peptide (CNP) is a well-characterized physiological meiotic inhibitor that has been successfully used to improve immature oocyte quality during in vitro maturation. However, the underlying roles of CNP in maternally aged oocytes have not been reported. Here, we found that the age-related reduction in the serum CNP concentration was highly correlated with decreased oocyte quality. Treatment with exogenous CNP promoted follicle growth and ovulation in aged mice and enhanced meiotic competency and fertilization ability. Interestingly, the cytoplasmic maturation of aged oocytes was thoroughly improved by CNP treatment, as assessed by spindle/chromosome morphology and redistribution of organelles (mitochondria, the endoplasmic reticulum, cortical granules, and the Golgi apparatus). CNP treatment also ameliorated DNA damage and apoptosis caused by ROS accumulation in aged oocytes. Importantly, oocyte RNA-seq revealed that the beneficial effect of CNP on aged oocytes was mediated by restoration of mitochondrial oxidative phosphorylation, eliminating excessive mitophagy. CNP reversed the defective phenotypes in aged oocytes by alleviating oxidative damage and suppressing excessive PINK1/Parkin-mediated mitophagy. Mechanistically, CNP functioned as a cAMP/PKA pathway modulator to decrease PINK1 stability and inhibit Parkin recruitment. In summary, our results demonstrated that CNP supplementation constitutes an alternative therapeutic approach for advanced maternal age-related oocyte deterioration and may improve the overall success rates of clinically assisted reproduction in older women.

## eLife assessment

This study presents **valuable** findings on the impact of C-natriuretic peptide (CNP) treatment in vivo on the fertility of aged mice. **Solid** data indicate CNP induces the cAMP-PKA pathway, causing reduced recruitment of Parkin protein to mitochondria in oocytes, resulting in reduced mitophagy, which may be significant for increased mitochondrial bioenergetics and improved cytoplasmic and nuclear maturation. The authors make additional claims regarding the mechanisms by which CNP impacts oocyte quality in vivo for which the evidence is inconclusive. This work will be of interest to reproductive biologists and clinical infertility specialists.

**\*For correspondence:**
weiq@nwsuaf.edu.cn (QW);
malab@nwafu.edu.cn (BM)

†These authors contributed equally to this work

**Competing interest:** The authors declare that no competing interests exist.

## Introduction

During growth, oocytes gradually acquire the capacity to resume meiosis, complete maturation, undergo successful fertilization, and achieve subsequent embryo developmental competence (*Gandolfi and Gandolfi, 2001*). However, ovarian involution precedes that of any other organ in female mammals, and in humans, the oocyte fertilization rate decreases rapidly after 35 years of age. Indeed, infertility associated with a decline in oocyte quality with increasing maternal age is a significant challenge.

Recently, with the development of cultural and social trends, many women have delayed childbearing, and ovarian senescence has become a public health problem (*Bartimaeus et al., 2020*; *Broekmans et al., 2009*). Ovarian aging is accompanied by abnormalities in organelle distribution, morphology, and functions, leading to inadequate oocyte growth, maturation, fertilization, and subsequent embryo development (*Reader et al., 2017*). Consequently, assisted reproductive technologies (ARTs) such as in vitro maturation (IVM) and in vitro fertilization (IVF) have become promising options for infertility treatment (*Chang et al., 2014*). However, further studies are needed to improve the subsequent developmental competence of maternally aged oocytes.

Oocyte maturation has two steps: nuclear maturation, which mainly involves germinal vesicle breakdown (GVBD) and chromosomal segregation, and cytoplasmic maturation, which involves redistribution of organelles (mitochondria, cortical granules [CGs], and the endoplasmic reticulum [ER]), changes in the intracellular ATP and antioxidant contents, and the accumulation of fertilization-related transcripts and proteins (*McClatchie et al., 2017*; *Watson, 2007*). The quality and developmental potential of aged oocytes are lower than those of oocytes derived from young females, primarily because aged oocytes exhibit negative consequences of cytoplasmic maturation, such as abnormal mitochondria and an aberrant CG distribution (*Miao et al., 2020*), deteriorated organelle and antioxidant system function, and increased reactive oxygen species (ROS) levels (*Zhang et al., 2019*). Excessive ROS generation leads to destructive effects on cellular components (*Zarkovic, 2020*). Notably, recent studies have indicated that increased oxidative damage is closely correlated with the occurrence of mitochondrial damage and mitophagy (*Jiang et al., 2021*), which is accompanied by blockade of oocyte meiosis (*Jin et al., 2022*; *Shen et al., 2021*).

The endogenous C-type natriuretic peptide (CNP) produced by follicular mural granulosa cells as a ligand of natriuretic peptide receptor 2 (NPR2), which is expressed primarily in cumulus cells, plays a crucial role in maintaining meiotic arrest (*Zhang et al., 2010*). Recent studies have suggested that CNP, an inhibitor of oocyte maturation, provides adequate time for cytoplasmic maturation, offering a new strategy to optimize the synchronization of nuclear and cytoplasmic maturation and improve the quality of immature oocytes in vitro (*Wei et al., 2017*). Moreover, CNP has been reported to enhance the antioxidant defense ability and developmental competence of oocytes in vitro (*Zhenwei and Xianhua, 2019*). Therefore, CNP may constitute a new alternative means to enhance antioxidant system function and protect against oxidative damage by eliminating excess ROS in aged oocytes. Although CNP has been suggested to contribute to improving the maturation and subsequent development of immature mouse oocytes in vitro, the effect of CNP on maternally aged oocytes remains to be determined.

In this study, we investigated the redistribution and function of organelles (mitochondria, CGs, and the ER), the ATP content, and the intracellular GSH level in CNP-treated maternally aged oocytes. The results showed that CNP improves the cytoplasmic maturation and developmental competence of maternally aged oocytes by optimizing organelle distribution and function and inhibiting PINK1/Parkin-mediated mitophagy. The findings of this study will contribute to understanding the mechanism of CNP in increasing the fertilization capacity and developmental ability of aged oocytes.

## Results

### CNP supplementation improves the quality of aged oocytes

To explore the effect of CNP on oocyte quality in aged mice, we first investigated whether intraperitoneal injection of CNP can affect oocyte quality. Young and aged mice were hormonally superovulated after 14 d of consecutive PBS or CNP daily injection (*Figure 1A and B*). As shown in *Figure 1C–E*, body weights were higher but ovary weight and the ratios of ovary to body weight were lower in the aged mice compared with their young counterparts. However, ovary weight and the ratios of ovary

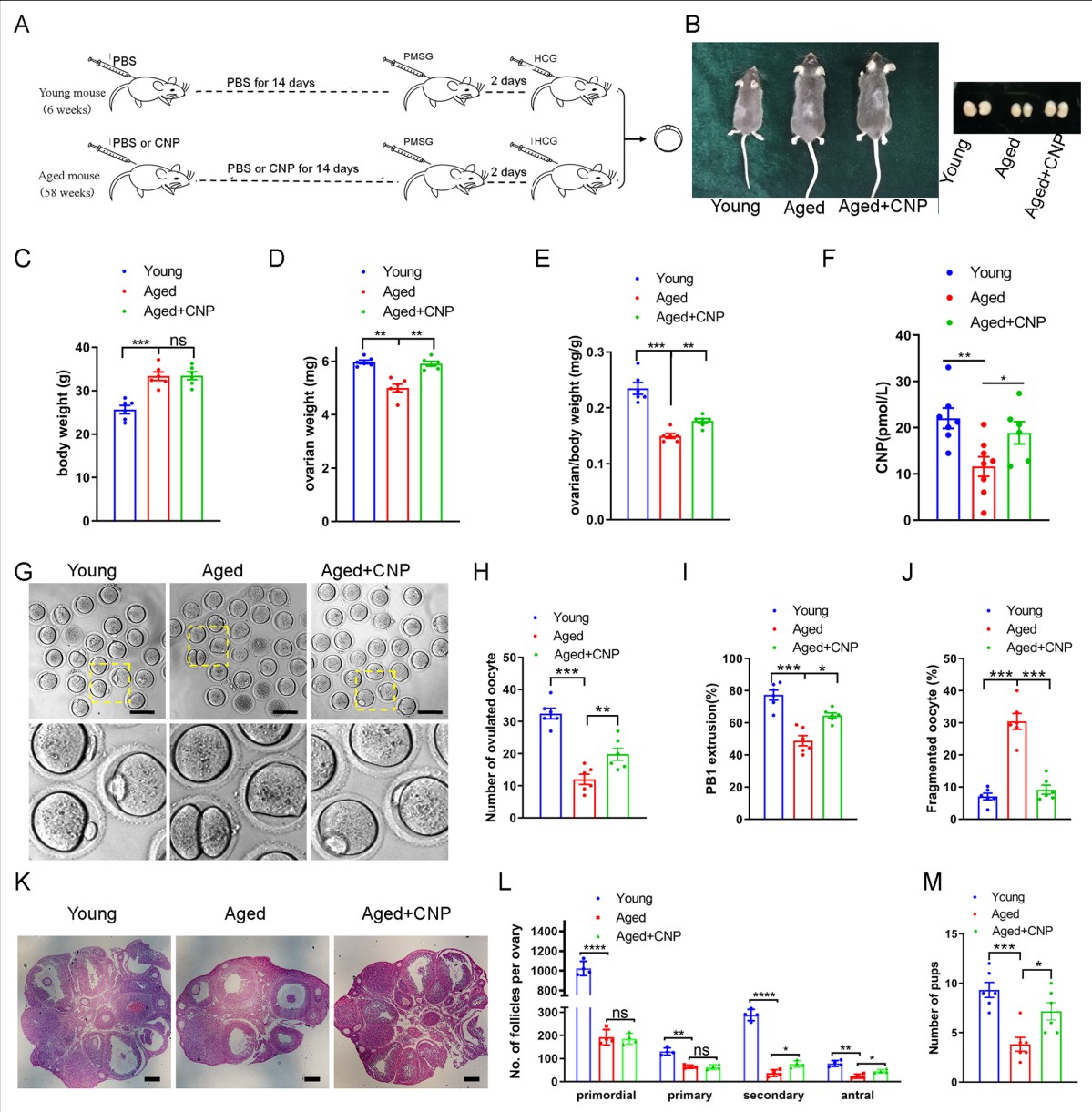

**Figure 1.** Effects of C-type natriuretic peptide (CNP) supplementation on the oocyte quality and female fertility in aged mice. (**A**) A timeline diagram of CNP administration and superovulation. (**B**) Representative images of young, aged, and CNP administration aged (Aged + CNP) mice as well as their ovaries. (**C**) Body weights of young, aged, and aged + CNP aged mice. (**D**) Ovarian weights of young, aged, and aged + CNP mice. (**E**) Ratios of ovarian weight to body weight for each group of mice. (**F**) Serum CNP concentrations were measured in young, aged, and aged + CNP mice. (**G**) Representative images of the oocyte polar body extrusion in young, aged, and aged + CNP mice. Scale bar: 100 µm. (**H**) Ovulated oocytes were counted in young, aged, and aged + CNP mice. (**I**) Rate of polar body extrusion in young, aged, and aged + CNP mice. (**J**) The rate of fragmented oocytes was recorded in young, aged, and aged + CNP mice. (**K**) Representative images of ovarian sections from young, aged, and aged + CNP mice. Scale bars: 100 µm. (**L**) Follicles at different developmental stages were counted in young, aged, and aged + CNP ovaries. (**M**) Average litter size of mated mice was assessed by mating with 2-month-old male mice.

The online version of this article includes the following figure supplement(s) for figure 1:

**Figure supplement 1.** Effects of C-type natriuretic peptide (CNP) on the maturation and spindle/chromosome structure in aged oocytes.

to body weight of the CNP-treated mice were significantly recovered (*Figure 1D and E*). Serum CNP concentrations were measured in young, aged, and aged + CNP-injected mice. The endogenous CNP content in serum from aged mice was markedly lower than that in serum from young mice (*Figure 1F*). In contrast, administration of CNP to aged mice significantly elevated the CNP content

in serum (*Figure 1F*). To further determine whether the elevated CNP content in serum can improve oocyte quality, we evaluated the number, first polar body (PB1) extrusion rate, and fragmentation rate. As the mice aged, the number of ovulations and the PB1 extrusion rate decreased significantly, but the incidence of fragmented oocytes increased dramatically (*Figure 1G–J*). Conversely, CNP supplementation apparently ameliorated the aging-induced defects in the number and morphology of the ovulated oocytes (*Figure 1G–J*). In addition, assessment of follicle development in the ovary sections by HE staining showed severe deterioration of follicles at different developmental stages in aged mice; however, CNP supplementation significantly increased the number of secondary follicles and antral follicles (*Figure 1K and L*). Young, untreated aged, and CNP-treated aged mice were naturally mated with 12-week-old male mice, and consistent with the increased number of ovulated oocytes with a normal morphology, the litter size of aged mice was also increased by CNP administration (*Figure 1M*).

To investigate the effects of CNP on IVM of aged mouse cumulus-oocyte complexes (COCs), we first examined the PB1 extrusion rate of COCs pretreated with 10 nM CNP to maintain meiotic arrest for 24 hr (pre-IVM) and then matured in vitro for 16 hr (a two-step culture system; *Figure 1—figure supplement 1A*). In the control (conventional IVM) group, only 35.36 ± 2.74% of the oocytes exhibited PB1 extrusion. After temporary meiotic arrest induced by treatment with 10 nM CNP, the maturation rate increased to 71.12 ± 3.02% (n = 104), significantly higher than that in the control group (p<0.01) (*Figure 1—figure supplement 1B and C*). The spindle morphology and chromosome alignment in in vitro-matured oocytes were also evaluated. The percentage of oocytes with abnormal spindle-chromosome complexes was significantly decreased in the group with 10 nM CNP-induced temporary meiotic arrest (*Figure 1—figure supplement 1D and E*). Collectively, these results indicate that CNP administration increased the serum CNP content, restored the number and morphology of aged oocytes, and improved the fertility of aged female mice.

## CNP supplementation restores cytoplasmic maturation events in maternally aged mouse oocytes

Pregnancy failure and fetal miscarriage increase with maternal age and, importantly, are associated with oocyte aneuploidy and spindle/chromosomal abnormalities (*Ma et al., 2020*). Therefore, we determined the rate of spindle/chromosomal abnormalities in oocytes of young, untreated aged, and CNP-treated aged mice by immunofluorescence staining and found that CNP treatment greatly improved the spindle/chromosomal abnormalities in aged mice (*Figure 2A and B*). To determine whether, in addition to affecting spindles/chromosomes, CNP supplementation affects other organelles during the maturation of aged oocytes, we examined the distribution of the Golgi apparatus, ER, and CGs in oocytes from young, untreated aged, and CNP-treated aged mice. The Golgi-Tracker results showed that in aged mouse oocytes the Golgi apparatus was distributed in agglutinated and clustered patterns, and CNP supplementation significantly reduced the rate of abnormal Golgi distribution (*Figure 2C and D*, *Figure 2—figure supplement 1A and B*). Since the ER plays an essential role in $Ca^{2+}$ signal-mediated oocyte fertilization and subsequent embryonic development (*Miyazaki and Ito, 2006*), we then examined the distribution pattern of the ER in oocytes. As shown in *Figure 2E*, the ER was accumulated at the chromosome periphery and was evenly distributed in the cytoplasm; however, the ER abnormally agglomerated in the cytoplasm and the chromosome periphery in a disorganized pattern in aged oocytes (*Figure 2E*, *Figure 2—figure supplement 2A*). Statistical analysis showed that the rate of abnormal ER distribution was significantly decreased in CNP-supplemented oocytes (*Figure 2F*, *Figure 2—figure supplement 2B*). The distribution of CGs is one of the most important indicators of oocyte cytoplasmic maturation and is related to the blockade of polyspermy following fertilization. We assessed whether CNP supplementation affects the distribution dynamics of CGs in aged oocytes. Lens culinaris agglutinin (LCA)-FITC staining showed that in young oocytes CGs were distributed evenly in the oocyte subcortical region, leaving a CG-free domain (CGFD) near chromosomes (*Figure 2G*). However, maternally aged oocytes showed an abnormal CG distribution, including increased migration of CGs towards the oocyte chromosomes or oocyte subcortical region, without leaving a CGFD (*Figure 2G and H*, *Figure 2—figure supplement 3A*). Consistent with this finding, statistical analysis of the fluorescence intensity of CG signals in aged oocytes showed a significant reduction compared with that in young oocytes, and CNP supplementation improved the mislocalization and decrease in the number of oocyte CGs (*Figure 2H and I*, *Figure 2—figure supplement*

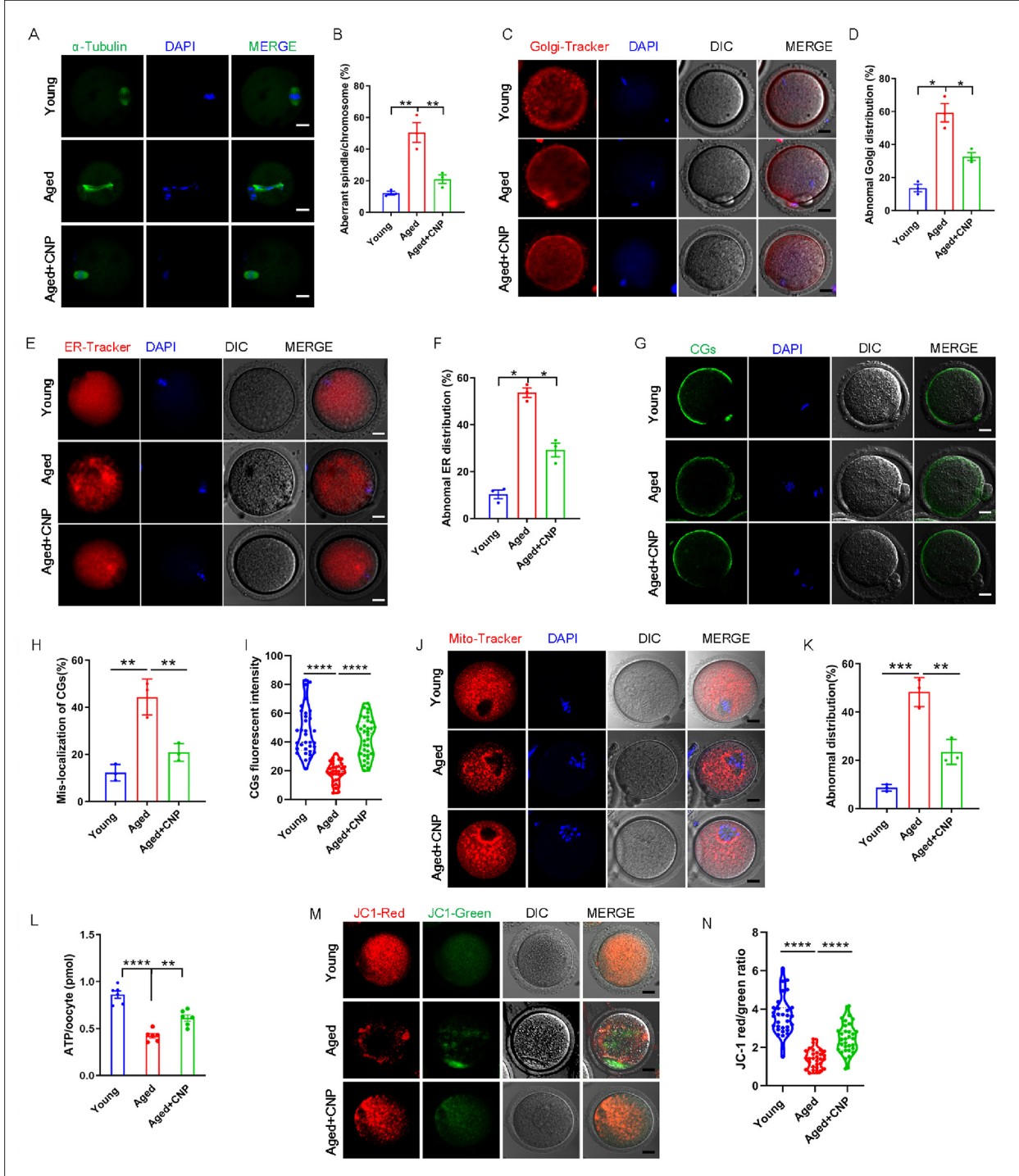

**Figure 2.** C-type natriuretic peptide (CNP) supplementation recovers cytoplasmic maturation events of maternally aged mouse oocytes. (**A**) Representative images of the spindle morphology and chromosome alignment at metaphase II in young, aged, and aged + CNP mice. Scale bar, 10 μm. (**B**) The rate of aberrant spindles at metaphase II was recorded in young, aged, and aged + CNP mice. (**C**) Representative images of the Golgi apparatus distribution at metaphase II in young, aged, and aged + CNP mice. Scale bar, 10 μm. (**D**) The rate of aberrant Golgi apparatus distribution was recorded in young, aged, and aged + CNP mice. (**E**) Representative images of the endoplasmic reticulum distribution at metaphase II in young, aged, and aged + CNP mice. Scale bar, 10 μm. (**F**) The rate of aberrant endoplasmic reticulum distribution was recorded in young, aged, and aged + CNP mice. (**G**) Representative images of the cortical granules (CGs) distribution in young, aged, and aged + CNP mice. Scale bar, 10 μm. (**H**) The rate of mislocalized CGs was recorded in the young, aged, and aged + CNP mice. (**I**) The fluorescence intensity of CG signals was measured in the young, aged, and aged + CNP mice oocyte. (**J**) Representative images of mitochondrial distribution in the young, aged, and aged + CNP mice oocytes stained with MitoTracker

*Figure 2 continued on next page*

*Figure 2 continued*

Red. Scale bar, 10 µm. (**K**) The abnormal rate of mitochondrial distribution was recorded in the young, aged, and aged + CNP mice oocytes. (**L**) ATP levels were measured in the young, aged, and aged + CNP mice. (**M**) Mitochondrial membrane potential (ΔΨm) was detected by JC-1 staining in the young, aged, and aged + CNP mice oocytes. Scale bar, 10 µm. (**N**) The ratio of red to green fluorescence intensity was calculated in the young, aged, and aged + CNP mice oocytes.

The online version of this article includes the following figure supplement(s) for figure 2:

**Figure supplement 1.** Effects of C-type natriuretic peptide (CNP) on Golgi apparatus distribution in aged oocytes.

**Figure supplement 2.** Effects of C-type natriuretic peptide (CNP) on endoplasmic reticulum distribution and function in aged oocytes.

**Figure supplement 3.** Effects of C-type natriuretic peptide (CNP) on the dynamics of cortical granules (CGs) in aged oocytes.

**Figure supplement 4.** Effects of C-type natriuretic peptide (CNP) on the mitochondrial distribution and function in aged oocytes.

*3B and C*). Taken together, these data imply that CNP is a potent agent for improving cytoplasmic maturation events in maternally aged mouse oocytes.

## CNP supplementation restores mitochondrial distribution and function in aged oocytes

To verify the effect of CNP supplementation on the mitochondrial distribution pattern and function in aged oocytes, we performed MitoTracker staining. In young oocytes, mitochondria exhibited a homogeneous distribution in the cytoplasm and accumulated at the periphery of chromosomes (*Figure 2J*). However, in aged oocytes, most mitochondria were aggregated in the cytoplasm and partially or completely failed to accumulate around chromosomes (*Figure 2J*, *Figure 2—figure supplement 4A*). Statistically, more than 40% of mitochondria in aged oocytes exhibited a mislocalized distribution pattern, and CNP supplementation significantly reduced the abnormal distribution rate (*Figure 2K*, *Figure 2—figure supplement 4B*). We then analyzed mitochondrial function by measuring the ATP content in oocytes from young, untreated aged, and CNP-treated aged mice. The ATP content in oocytes from aged mice was considerably lower than that in oocytes from young mice but was restored following CNP supplementation (*Figure 2L*, *Figure 2—figure supplement 4C*). We also tested the mitochondrial membrane potential, which has been shown to be the driving force of mitochondrial ATP synthesis, by staining with the potentiometric dye JC-1 (*Figure 2M*, *Figure 2—figure supplement 4D*). The mitochondrial membrane potential was lower in oocytes from aged mice than in oocytes from young mice but was restored in oocytes from CNP-supplemented aged mice (*Figure 2M and N*, *Figure 2—figure supplement 4D and E*). Overall, these observations suggest that CNP supplementation improved aging-induced mitochondrial dysfunction in oocytes.

## CNP supplementation eliminates excessive ROS and attenuates DNA damage and apoptosis in aged oocytes

We proposed that mitochondrial dysfunction induces ROS imbalance and oxidative stress in aged oocytes. To test this hypothesis, we carried out dichlorofluorescein (DCFH) staining to measure ROS levels in each group of oocytes (*Figure 3A*). Quantitative analysis of the fluorescence intensity showed that ROS signals were markedly enhanced in aged oocytes compared with young oocytes (*Figure 3B*). Conversely, CNP supplementation effectively reduced the ROS accumulation observed in aged oocytes (*Figure 3A and B*, *Figure 3—figure supplement 1A and B*). In addition to being caused by ROS accumulation, age-associated oxidative stress damage can be caused by reduced antioxidant defense system function. We therefore investigated whether CNP contributes to improving the antioxidant defense ability in aged oocytes. Quantification of the nicotinamide adenine dinucleotide phosphate (NADPH) levels and the ratio of reduced to oxidized glutathione (GSH/GSSG ratio) in oocytes showed that NADPH levels and the GSH/GSSG ratio were decreased in oocytes from aged mice compared with those from young mice and that CNP treatment significantly increased NADPH levels and the GSH/GSSG ratio in oocytes from aged animals (*Figure 3C and D*). Because a high level of ROS not only results in the accumulation of DNA damage but also causes oocyte apoptosis (*Miao et al., 2020*), we next evaluated DNA damage and apoptosis in oocytes by γ-H2A.X and Annexin-V staining, respectively. As expected, higher signals indicating DNA damage and apoptosis were observed in aged oocytes than in young oocytes, and these increases were alleviated by supplementation with CNP (*Figure 3E–H*). Taken together, these observations suggested that the rates of

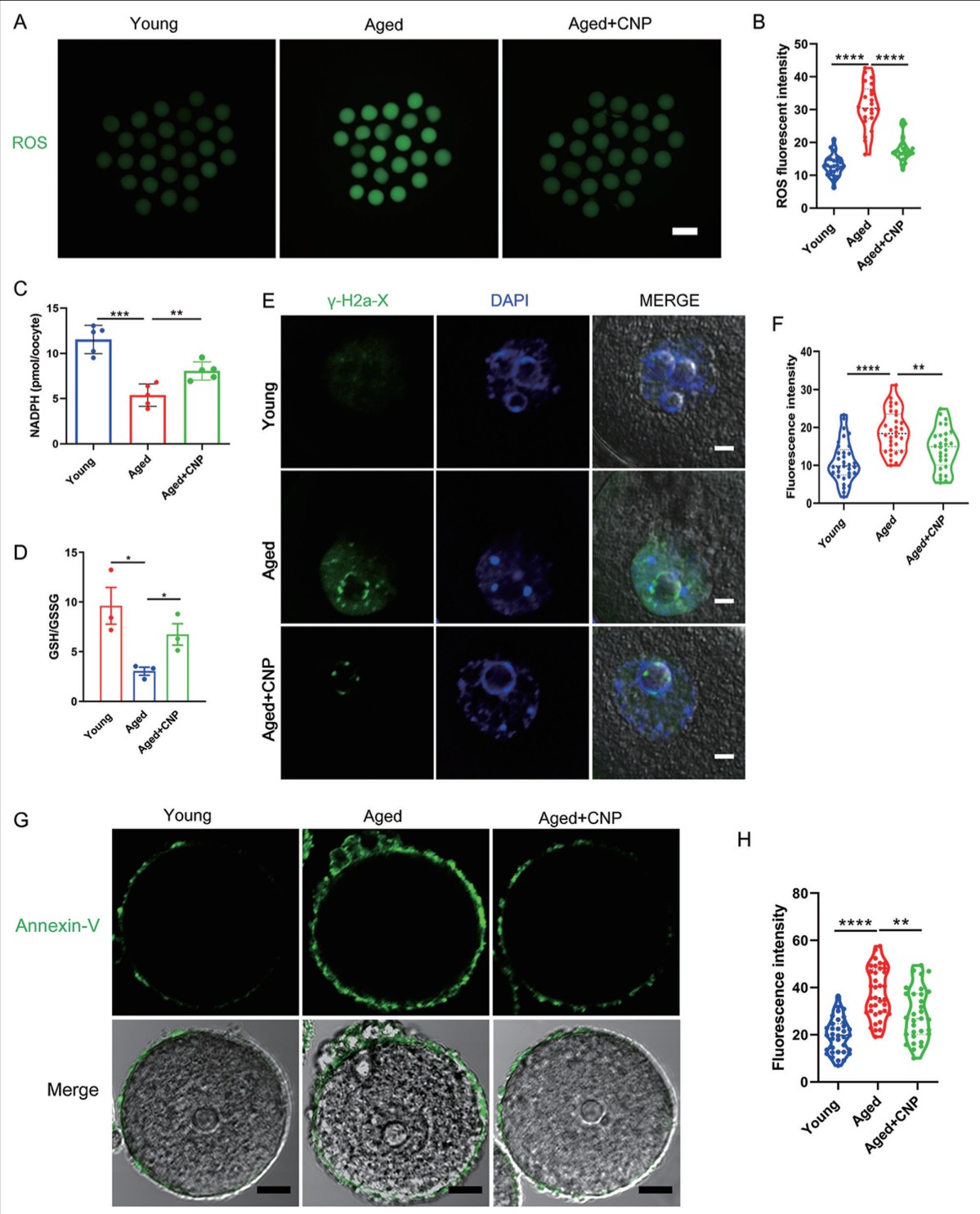

**Figure 3.** Effects of C-type natriuretic peptide (CNP) on the reactive oxygen species (ROS) content, DNA damage, and apoptosis in aged oocytes.
(**A**) Representative images of ROS levels detected by dichlorofluorescein (DCFH) staining in the young, aged, and aged + CNP mice oocytes. Scale bar, 100 μm. (**B**) The fluorescence intensity of ROS signals was measured in the young, aged, and aged + CNP mice oocytes. (**C**) Oocyte NADPH levels in the young, aged, and aged + CNP mice were measured. (**D**) The ratio of GSH/GSSG was measured in the young, aged, and aged + CNP mice oocytes. (**E**) Representative images of DNA damage stained. with the γ-H2AX antibody in young, aged, and aged + CNP oocytes. Scale bar, 10 μm. (**F**) γ-H2AX fluorescence intensity was counted in young, aged, and aged + CNP oocytes. (**G**) Representative images of apoptotic status, assessed by Annexin-V staining, in young, aged, and aged + CNP oocytes. Scale bar, 20 μm. (**H**) The fluorescence intensity of Annexin-V signals was measured in young, aged, and aged + CNP oocytes.

*Figure 3 continued on next page*

*Figure 3 continued*

The online version of this article includes the following figure supplement(s) for figure 3:

**Figure supplement 1.** Effects of C-type natriuretic peptide (CNP) on the reactive oxygen species (ROS) content in aged oocytes.

DNA damage and apoptosis are higher in aged oocytes, possibly because of maternal aging-induced excessive accumulation of ROS. Notably, our results demonstrated that CNP supplementation exerts antioxidant activity, which is an effective strategy to ameliorate maternal aging-induced DNA damage and apoptosis in oocytes.

## CNP supplementation improves the fertilization ability and early embryo development of aged oocytes

Considering that oocyte fertilization and subsequent embryo developmental competence are profoundly affected by mitochondrial function, we then tested whether the oocyte fertilization capacity and normal development to the blastocyst stage are enhanced by CNP. The IVF results showed that aged oocytes had dramatically lower fertilization rates than young oocytes and that CNP supplementation effectively increased the fertilization rate of aged oocytes (*Figure 4A and B*). We further examined the subsequent developmental ability of the fertilized oocytes. As expected, CNP supplementation effectively increased the blastocyst formation rate of aged oocytes both in vivo (*Figure 4A,*

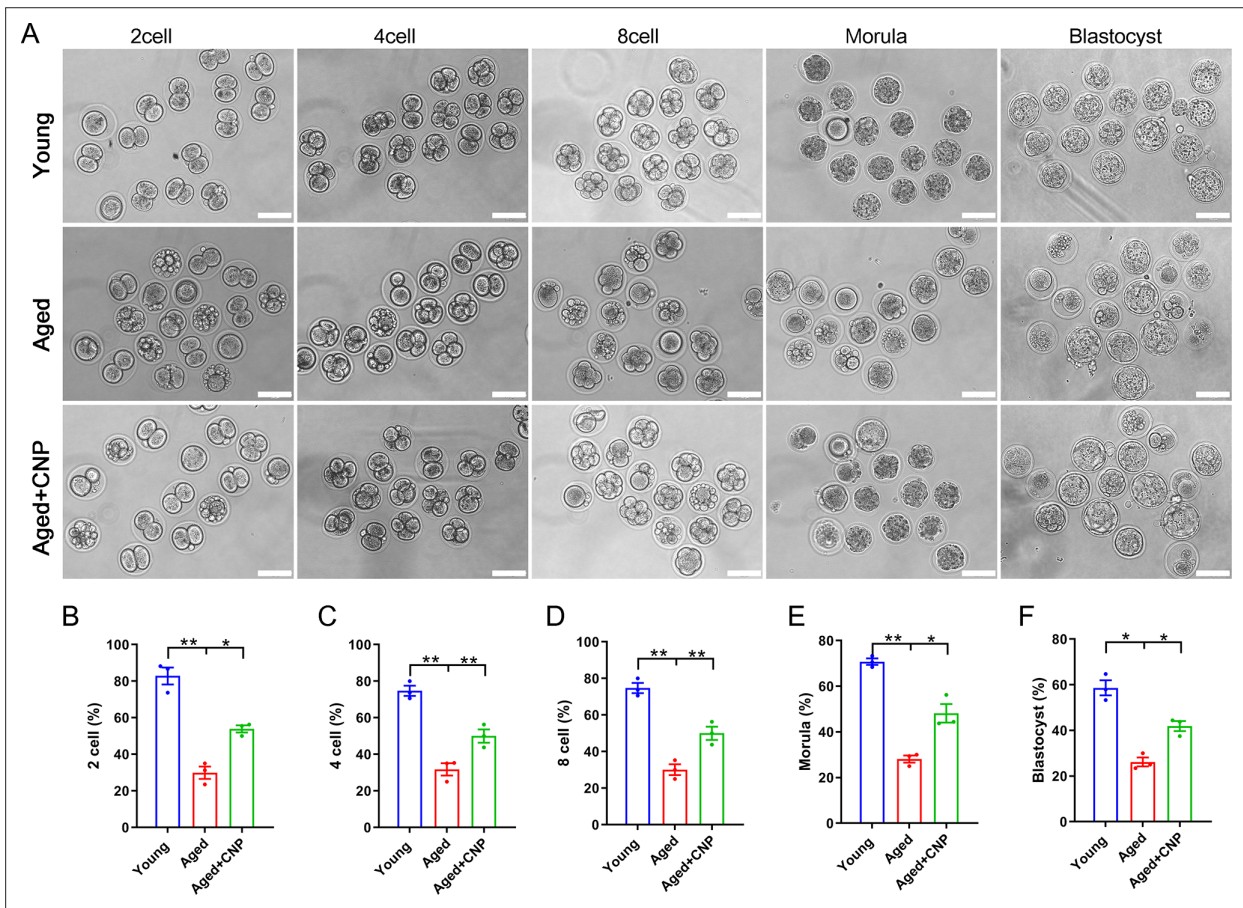

**Figure 4.** Effects of C-type natriuretic peptide (CNP) on the fertilization ability and embryonic development of aged oocytes. (**A**) Representative images of early embryos developed from young, aged, and aged + CNP oocytes in vitro fertilization. Scale bar, 100 µm. (**B**) The fertilization rate (two-cell embryos rate), (**C**) four-cell embryos rate, (**D**) eight-cell embryos rate, (**E**) morula rate, and (**F**) blastocyst formation rates were recorded in the young, aged, and aged + CNP groups. Data in (**B**–**F**) are presented as mean percentage (mean ± SEM) of at least three independent experiments.

The online version of this article includes the following figure supplement(s) for figure 4:

**Figure supplement 1.** Effects of C-type natriuretic peptide (CNP) on the fertilization ability and embryonic development in vitro maturation oocytes.

C–F) and in vitro (*Figure 4—figure supplement 1*). These results demonstrate that CNP increases the fertilization capacity and promotes subsequent embryonic development of oocytes from aged mice.

## Identification of target effectors of CNP in aged oocytes by single-cell transcriptome analysis

To verify the cellular and molecular mechanisms of CNP supplementation in improving oocyte quality in aged mice, we performed single-cell transcriptome analysis of GV oocytes derived from young, untreated aged, and CNP-treated aged mice to identify potential target effectors. The relative expression of several randomly selected genes from each group was verified using quantitative real-time PCR (*Figure 5—figure supplement 1A and B*). As shown in the heatmap and volcano plot, the transcriptome profile of aged oocytes was significantly different from that of young oocytes, with 77 differentially expressed genes (DEGs) downregulated and 440 DEGs upregulated in aged oocytes identified through DEGseq2 analysis (*Figure 5A–C*). Furthermore, CNP supplementation resulted in downregulation of 584 genes and upregulation of 527 genes compared with aged oocytes (*Figure 5—source data 1*). In particular, Kyoto Encyclopedia of Genes and Genomes (KEGG) enrichment analysis showed that genes enriched in the ubiquitin-mediated proteolysis and mitophagy pathways were abnormally highly expressed in aged oocytes compared with young oocytes but that these expression levels were restored to the baseline levels in CNP-supplemented aged oocytes (*Figure 5D and E*). In addition, oxidative phosphorylation and peroxisome proliferator-activated receptor (PPAR) signaling pathways were ranked at the top of the enrichment list in CNP-supplemented aged oocytes compared to untreated aged oocytes, consistent with our abovementioned observations that CNP supplementation improved mitochondrial function in aged oocytes. Many of the enriched KEGG enrichment pathways are highly related to mitophagy and mitochondrial function, which suggests that mitophagy should be strongly considered as a CNP effector in aged oocytes.

## CNP supplementation attenuates oxidative damage by inhibiting mitophagy in aged oocytes

To verify the effect of CNP supplementation on mitophagy in aged oocytes, we first analyzed mitochondrial structure in the oocytes of young, untreated aged, and CNP-treated aged mice by transmission electron microscopy (*Figure 6A*). Quantitatively, mitochondrial damage, as evidenced by membrane rupture and a lack of electron density, was significantly increased in aged oocytes but was ameliorated in CNP-supplemented aged oocytes (*Figure 6B*). Because oxidative stress has been implicated in triggering mitochondrial oxidative injury and mitophagy (*Adhikari et al., 2022*; *Shen et al., 2016*), we next determined whether supplementation with CNP can eliminate excessive mitochondrial ROS (mtROS). As expected, supplementation with CNP substantially reduced the mtROS signals, as shown by MitoSOX staining and fluorescence intensity measurements (*Figure 6C and D*). We then evaluated degradation of the autophagy biomarker p62, the accumulation of LC3-II, the conversion of LC3-I to LC3-II, and the expression patterns of the mitophagy-related proteins PINK1 and Parkin (*Figure 6E*). Western blot analysis revealed that aged oocytes exhibited significant p62 degradation, LC3-II accumulation, and marked increases in PINK1 and Parkin expression levels, whereas CNP supplementation abrogated these effects (*Figure 6E–I*). Collectively, the above data indicate the inhibitory effect of CNP on oocyte mitophagy through the PINK1-parkin signaling pathway.

## CNP downregulates Parkin recruitment and mitophagy via the cAMP/PKA pathway

How PINK1- and Parkin-mediated mitophagy is regulated by CNP in aged oocytes, however, requires further elucidation. The cAMP/PKA signaling pathway, which is dependent on the phosphorylation of mitochondrial proteins, has emerged as a direct means to regulate mitophagy and mitochondrial physiology (*Ould Amer and Hebert-Chatelain, 2018*; *Lobo et al., 2020*). The concentrations of cAMP in GV oocytes derived from young, untreated aged, and CNP-treated aged mice were determined. As shown in *Figure 6J*, the cAMP concentration in aged oocytes was significantly lower than that in young oocytes, but administration of CNP resulted in a substantial increase in intraoocyte cAMP concentrations. This increase in cAMP significantly reduced mitochondrial recruitment of Parkin and mitophagy, which were dependent on PKA activity (*Lobo et al., 2020*). Next, we applied a PKA inhibitor, H89, to determine whether PKA is directly involved in CNP-mediated oocyte mitophagy.

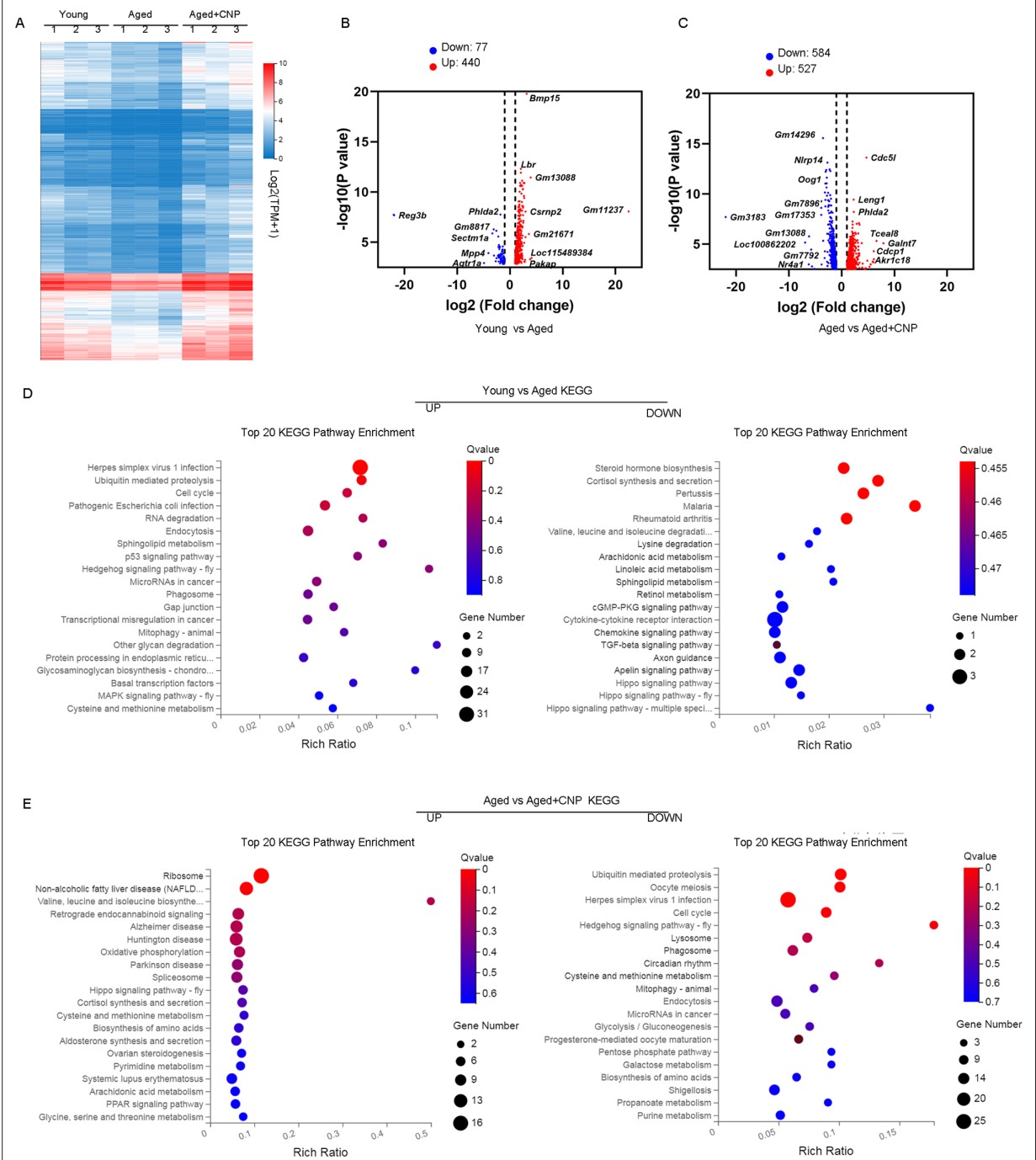

**Figure 5.** Effect of C-type natriuretic peptide (CNP) supplementation on transcriptome profiling of aged oocytes. (**A**) Heatmap illustration displaying gene expression of young, aged, and aged + CNP oocytes. (**B**) Volcano plot showing differentially expressed genes (DEGs; downregulated, blue; upregulated, red) in young vs. aged oocytes. Some highly DEGs are listed. (**C**) Volcano plot showing DEGs in aged vs. aged + CNP oocytes. Some highly DEGs are listed. (**D**) Kyoto Encyclopedia of Genes and Genomes (KEGG) enrichment analysis of upregulated and downregulated DEGs in young vs. aged oocytes. (**E**) KEGG enrichment analysis of upregulated and downregulated DEGs in aged vs. aged + CNP oocytes.

The online version of this article includes the following source data and figure supplement(s) for figure 5:

**Source data 1.** RNAseq dataset.

**Figure supplement 1.** The relative expression of several randomly selected genes from each group was verified using quantitative real-time PCR.

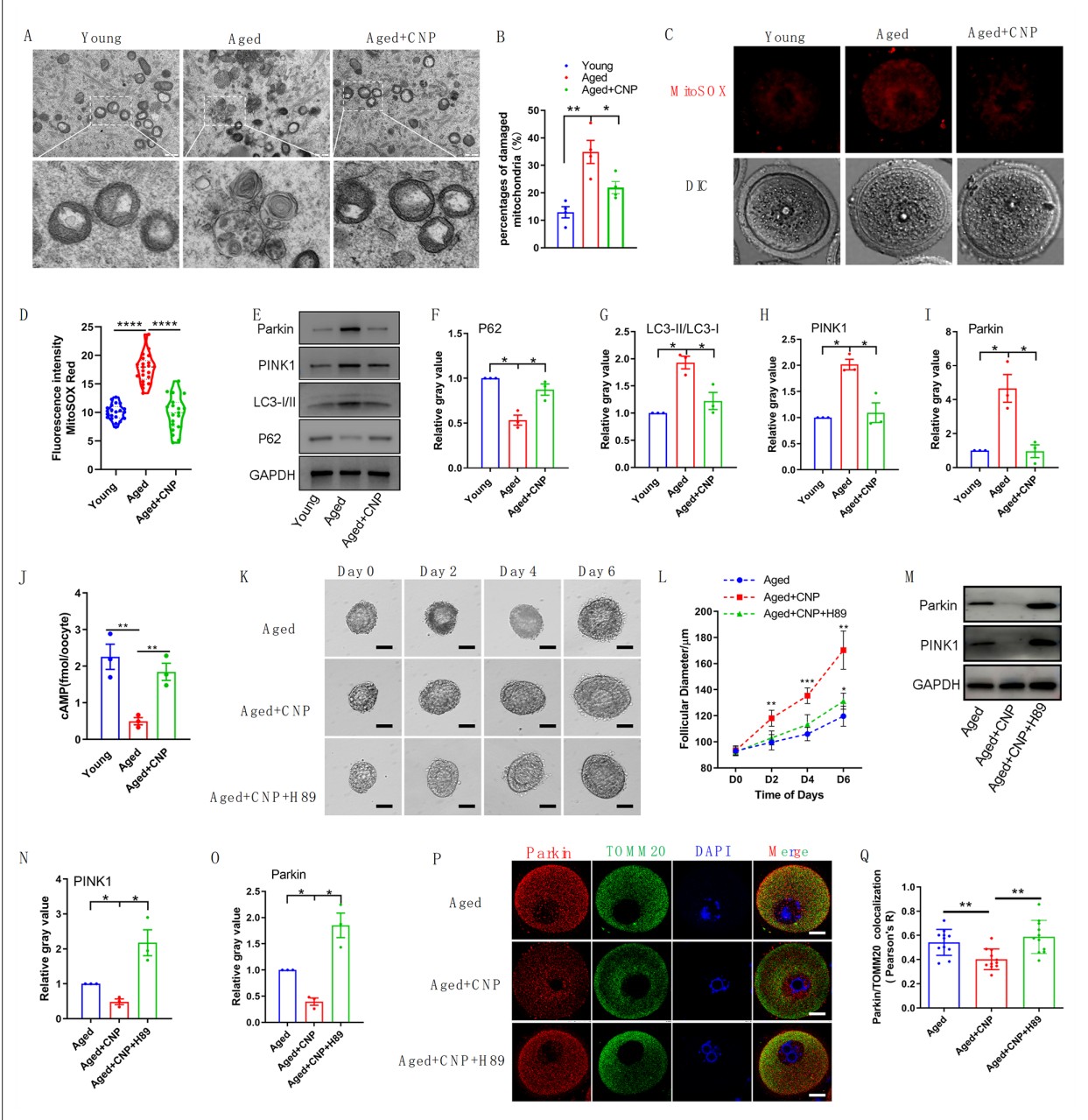

**Figure 6.** Evaluation of C-type natriuretic peptide (CNP) supplementation on mitophagy activity of aged oocytes. (**A**) Representative images of mitochondria morphology and structure in young, aged, and aged + CNP oocytes by TEM. (**B**) Accumulation of mitochondria damage in young, aged, and aged + CNP oocytes. Under TEM images, percentages of damaged mitochondria per area (500 nm × 500 nm) were shown. At least four visions were chosen and mitochondria were counted by two individuals. (**C**) Representative images of mitochondria reactive oxygen species (ROS) stained with MitoSOX in young, aged, aged + CNP oocytes. Scale bar, 20 μm. (**D**) Fluorescence intensity of MitoSOX signals was measured in young aged, aged + CNP oocytes. (**E**) Western blots of P62（62 kDa）, LC3-I/II (14-16 kDa), PINK1（60 kDa）, and Parkin（50 kDa）in young, aged, and aged + CNP oocytes. GAPDH (37 kDa) was used as internal control. (**F–I**) Relative gray value of proteins detected with western blots compared with controls. (**J**) Oocyte cAMP concentrations were measured in young, aged, and aged + CNP mice. (**K**) Representative images at day 0, day 2, day 4, and day 6 of cultured preantral follicles with or without CNP or CNP + H89 treatment. Scale bar = 50 μm. (**L**) Diameters of preantral follicles with or without CNP or CNP + H89 treatment from day 0 to day 6. Six independent culture experiments were performed. (**M**) Western blots of PINK1 and Parkin in aged, aged + CNP, and aged + CNP + H89-treated oocytes. GAPDH was used as internal control. (**N–O**) Relative gray value of proteins detected with western blots compared with controls. (**P**) Double immunofluorescence staining of Parkin and TOMM20. The mitochondria outer membrane protein TOMM20 was performed to reveal the translocation of PRKN proteins on mitochondria. Red, PRKN; green, TOMM20; blue, DNA was labeled with Hoechst 33342. Bar: 20 μm. (**Q**) The colocalization of Parkin and TOMM20 in oocytes from aged, aged + CNP, and aged + CNP + H89-treated mice was compared. Pearson's R shows the results of co-location analysis.

We isolated preantral follicles (80–100 μm diameter) from the ovaries of aged mice and treated them with 100 nM CNP or 100 nM CNP + 10 μM H89 during in vitro culture. Monitoring of follicle growth dynamics showed that treatment with 100 nM CNP significantly increased the follicle diameter (*Figure 6K and L*), whereas H89 treatment inhibited the effect of CNP on promoting preantral follicle growth (*Figure 6K and L*). Western blot analysis revealed that CNP supplementation significantly decreased PINK1 and Parkin expression levels, but H89 treatment abrogated these expression changes (*Figure 6M–O*). The cAMP-PKA pathway plays an important role in inhibiting Parkin recruitment to damaged mitochondria (*Akabane et al., 2016*). We therefore sought to determine whether PKA inhibition regulates Parkin recruitment. The effects of CNP on mitochondria were examined by double staining for Parkin and translocase of outer mitochondrial membrane 20 (TOMM20). CNP clearly inhibited the mitochondrial localization of Parkin, but inhibition of PKA with H89 resulted in Parkin translocation to mitochondria, as shown by the overlap of the two staining signals (*Figure 6P and Q*). Collectively, these data suggested that the suppression of Parkin recruitment through the cAMP-PKA axis is an important mechanism underlying the protective effect of CNP against oxidative injury in maternally aged mouse oocytes.

## Discussion

In mammals, the endogenous peptide CNP is expressed by endothelial cells in many tissues and has diverse physiological functions in mediating cardioprotective effects, bone growth, oocyte meiotic progression, and follicle growth and development (*Bae et al., 2017*; *Moyes and Hobbs, 2019*; *Peake et al., 2014*; *Sato et al., 2012*; *Xi et al., 2019*). Beyond the role of CNP as an oocyte meiotic arrest factor, previous studies by our group and others confirmed that adding CNP to the pre-IVM system significantly improved oocyte maturation and subsequent embryo developmental potential (*Richani and Gilchrist, 2022*; *Soto-Heras et al., 2019*; *Wei et al., 2017*). The synchronization of nuclear and cytoplasmic maturation is essential for oocyte quality and supporting early embryonic preimplantation development. However, the underlying molecular mechanism and whether CNP has any beneficial effect on the maternal age-induced decline in oocyte quality are incompletely understood. In this study, we showed that CNP levels declined with age and demonstrated that CNP supplementation increased the number of antral follicles and the ovulation rate and enhanced oocyte quality and fertility. Furthermore, supplementation of CNP in pre-IVM oocyte culture medium reversed the adverse effects of age on immature oocytes, offering a potentially effective approach for ARTs to acquire a greater number of high-quality oocytes and improve the fertility of older women.

Many factors affect the adverse effects on the oocyte maturation process and embryonic development associated with advanced maternal age (*Mikwar et al., 2020*). ARTs are an efficient scheme to resolve infertility as maternal fertility declines with aging. However, the low success rate of IVM oocytes, which is especially pronounced in maternally aged oocytes, limits fertilization outcomes. Our in vivo results showed that CNP supplementation results in multiple improvements, including reductions in oxidative damage, spindle defects, and abnormal organelle distributions and functions, in maternally aged oocytes. Thus, we further investigated the use of CNP in the IVM system, especially in improving the quality of oocytes derived from aged mice. The results indicated that CNP-induced temporary meiotic arrest improved the maturation and fertilization rate of maternally aged oocytes and increased their subsequent embryo developmental competence. Specifically, our results confirmed an advanced role for CNP in preventing the development of mitochondrial structure abnormalities and the typical dysfunctional processes in aged oocytes. Furthermore, these data showed that CNP apparently improved the antioxidant defense system impairment accompanying oocyte aging and alleviated oxidative stress. In addition, the findings demonstrated that CNP improved cytoplasmic maturation events by maintaining normal CG, ER, and Golgi apparatus distribution and mitochondria function in aged oocytes.

The asynchronous nature of nuclear and cytoplasmic maturation is a major challenge in improving the quality of IVM oocytes (*Coticchio et al., 2015*). Age-related aberrant chromosome alignment prior to cytoplasmic maturation may result in poor oocyte quality and subsequent reduced reproductive potential (*Russ et al., 2022*). In this study, after COCs were induced, the temporary meiotic arrest resulting from CNP treatment significantly increased the maturation rate, which may synchronize oocyte nuclear and cytoplasmic maturation. Organelle distribution is a necessary feature of oocyte cytoplasmic maturation and subsequent development. Dramatic ER reorganization (*FitzHarris et al.,*

*2007*), CG translocation to the cell cortex (*Liu et al., 2003*), and the Golgi apparatus distribution and function are commonly regarded as indicators of cytoplasmic maturation (*Mao et al., 2014*). Similarly, our results demonstrate that CNP supplementation restored cytoplasmic maturation in maternally aged oocytes by ensuring normal organelle distribution dynamics and organelle function, increasing the fertilization capacity and developmental competence of aged oocytes. It is reasonable to assume that CNP is a potential option to prevent abnormal organelle distribution and functions in oocytes that could be triggered by ubiquitous environmental endocrine disruptors, such as bisphenol A and citrinin (*Pan et al., 2021*; *Sun et al., 2020*). CGs are oocyte-specific vesicles located under the subcortex. Fusion of CGs with the oocyte plasma membrane is the most important event needed to prevent polyspermy (*Miao et al., 2020*). The distribution of CGs is usually regarded as one of the most important indicators of oocyte cytoplasmic maturation. The contents of the CGs are normally discharged by exocytosis when the egg is stimulated by the fertilizing spermatozoon; this process is called the cortical reaction, and it prevents polyspermy and protects the embryo from a hostile environment during early development (*Schuel, 1978*).

One of the major known causes of oocyte oxidative damage and apoptosis arises from excessive ROS accumulation with aging, especially in IVM oocytes (*Combelles et al., 2009*; *Soto-Heras and Paramio, 2020*).Excessive ROS accumulation occurs as a result of two processes, namely, constant generation in the mitochondria or scavenging by antioxidant defense systems, both of which involve age-related quality decreases in oocytes (*Zhang et al., 2020a*). Thus, maintaining the balance between the production and scavenging of ROS could help to alleviate age-related oxidative damage and fertility decreases. Some antioxidative factor(s) within oocytes might deteriorate as the potential mother ages, compromising the ability for ROS scavenging (*Schwarzer et al., 2014*). GSH serves as one of the antioxidants in oocytes to combat ROS-mediated oxidative stress, which is highly correlated with oocyte developmental competence (*Furnus et al., 2008*). The present results suggested that CNP-induced temporary meiotic arrest increased the GSH/GSSG ratio, which is involved in the enhancement of oocyte antioxidant defense and may contribute to improving oocyte developmental competence. Consistent with previous studies (*Miao et al., 2020*), our findings validated that maternal aging results in excessive accumulation of ROS and DNA damage, which severely impairs follicle development, ovulation, oocyte quality and subsequent embryo developmental potential.

Defects in chromosome separation and decondensation as well as chromosomal misalignment caused by spindle detachment are the major contributing factors responsible for the decline in oocyte quality with aging (*Chiang et al., 2011*; *Eichenlaub-Ritter et al., 2004*). Oocytes require ATP for spindle formation, chromosome segregation, and polar body extrusion and fertilization processes (*Arhin et al., 2018*; *Eichenlaub-Ritter, 2012*). Mitochondria are the most abundant organelles in oocytes and play an important role in ATP production via oxidative phosphorylation to phosphorylate adenosine diphosphate (*Bentov et al., 2011*). Thus, mitochondrial function is a key indicator of oocyte quality and successful fertilization in ARTs (*Mikwar et al., 2020*). Mitochondrial metabolic activity and mitochondrial DNA replication dramatically decrease in oocytes with maternal age, which reduces ATP production; leads to meiotic spindle damage, chromosome misalignment, and aneuploidy; and largely impairs oocyte maturation processes (*Eichenlaub-Ritter et al., 2011*; *May-Panloup et al., 2016*). We demonstrated that CNP reverses mitochondrial dysfunction induced by aeing in oocytes by analyzing the mitochondrial distribution, ATP content, and mitochondrial membrane potential ($\Delta \Psi$m).

Notably, disruption of the mitochondrial membrane potential is a potent trigger of mitophagy (*Matsuda et al., 2010*). Our single-cell transcriptome profiling data showed that the expression of genes related to ubiquitin-mediated proteolysis and the mitophagy pathway was considerably upregulated in aged oocytes but restored to normal levels following CNP supplementation. We also observed by TEM that CNP supplementation suppressed the accumulation of autophagic vesicles containing mitochondria. Furthermore, immunoblot analysis revealed degradation of the autophagy biomarker p62 and accumulation of LC3-II in aged oocytes, events that were markedly suppressed in CNP-treated oocytes. In general, excessive activation of mitophagy and mitochondrial damage in aged oocytes may be involved in the deterioration of oocyte quality, while CNP can ameliorate this process.

The PINK1/Parkin pathway is one of the most studied ubiquitin-dependent mitophagy processes and is crucial for the equilibrium between mitochondrial biogenesis and mitochondrial removal via selective recognition and elimination of dysfunctional mitochondria (*Vernucci et al., 2019*). In healthy

mitochondria, the serine/threonine kinase PINK1 is usually expressed at low levels, but it rapidly accumulates on damaged or aged mitochondria that exhibit loss of membrane potential (*Narendra and Youle, 2011*). The decrease in the mitochondrial membrane potential abolishes translocation across the outer and inner membranes and confines PINK1 in the mitochondrial matrix, stabilizing it on the mitochondrial outer membrane in a complex with the translocase TOM (*De Gaetano et al., 2021*). Stabilized PINK1 recruits the cytosolic E3-ubiquitin ligase Parkin from the cytosol to damaged mitochondria, an event followed by mitophagy. The effects of CNP on meiotic arrest depend on the maintenance of cAMP levels in oocytes (*Zhang et al., 2010*). Recent findings revealed that cAMP-dependent activation of PKA reduced the PINK1 protein level due to its rapid degradation via the proteasome and severely inhibited Parkin recruitment to depolarized mitochondria (*Lobo et al., 2020*). Our data confirmed that PINK1 expression was decreased in CNP-treated oocytes, thus leading to a reduction in Parkin recruitment. However, these effects were disrupted by the inhibition of PKA pathway activity, indicating that cAMP indeed mediates the ameliorating effects of CNP on aged oocyte quality. Taken together, these findings indicate that PKA-mediated inhibition of Parkin recruitment may contribute to protecting mitochondria with a low membrane potential from mitophagy in aged oocytes.

Collectively, our studies demonstrated that CNP improves the fertilization and developmental competence of maternally aged mouse oocytes by preventing age-related antioxidant defects and excessive mitophagy. Considering its known contributions as a physiological meiotic inhibitor, CNP provides an alternative to prevent maternal age-related oocyte quality defects and improve developmental competence. Although ARTs have been widely used to treat infertility, their overall success rates in women of advanced maternal age remain low. Our data may provide a new theoretical basis for the use of CNP in improving subfertility in older women or the application of clinically assisted reproduction. Out of caution, however, randomized controlled clinical trials should be conducted to further study the efficacy of CNP in women who wish to become pregnant.

## Materials and methods

### Animals and ethics statement

The young (6–8-week-old) and aged (58–60-week-old) C57BL/6J female mice were obtained from the Experimental Animal Center of the Xi'an Jiaotong University and housed in a temperature- (20–25°C) and light-controlled environment (12 hr light–12 hr dark cycle) and provided with food and water ad libitum.

### In vivo treatment with CNP

Aged mice (58-week-old) were intraperitoneally injected daily with CNP (120 µg/kg body weight; Cat#B5441, ApexBio) for 14 d. CNP was dissolved in PBS and diluted to appropriate concentration by physiological saline solution before injection. The mice were followed by a single injection of 5 IU pregnant mare serum gonadotropin (PMSG; Ningbo Second Hormone Factory, Ningbo, China) for 46 hr to stimulate penultimate follicle maturation before collection of ovaries for histological analyses and weighting. Some mice were further injected with 5 IU human chorionic gonadotropin (hCG; Ningbo Second Hormone Factory), and the ovulated oocytes in oviducts were monitored 16 hr later to evaluate ovulation efficiency.

### Measurement of CNP levels

CNP was measured in plasma by a two-site polyclonal direct ELISA kit (Biomedica Medizinprodukte, Vienna, Austria) according to the manufacturer's instructions. Collect blood samples in standardized serum separator tubes (SST), allow samples to clot for 30 min at room temperature, and perform serum separation by centrifugation. Assay the acquired serum samples immediately. Read the optical density (OD) of all wells on a plate reader using 450 nm wavelength. Construct a standard curve from the absorbance read-outs of the standards. Obtain sample concentrations from the standard curve.

### Histological analysis of ovaries

Ovaries from each group of mice were fixed in 4% paraformaldehyde (pH 7.5) overnight at 4°C, dehydrated using graded ethanol, followed by xylenes, and embedded in paraffin. Paraffin-embedded

ovaries were serial sectioned at a thickness of 5 µm for hematoxylin and eosin (H&E) staining. Ovaries from three mice of each group were used for the analysis.

## Collection and culture of COCs

Female mice were stimulated by an intraperitoneal injection of 5 IU PMSG, and mice were sacrificed by cervical dislocation 24 hr later. The ovaries were collected, and the well-developed Graafian follicles were punctured with 30-gauge needles to collect COCs. Only COCs with morphological integrity and a distinct germinal vesicle (GV) were cultured in basic culture medium consisted of Minimum Essential Medium (MEM)-α (Life Technologies, New York) supplemented with 3 mg/mL bovine serum albumin and 0.23 mM pyruvate at 37°C under an atmosphere of 5% $CO_2$ in air with maximum humidity.

## CNP treatment and in vitro maturation

For in vitro induce temporary meiotic arrest, COCs were cultured in basic culture medium containing 10 nM of CNP. The dose of CNP for meiotic arrest in mouse oocytes in vitro was selected based on the published literatures (*Zhang et al., 2011*) and our preliminary reports (*Wei et al., 2017*). After meiotic arrest culture for 24 hr, COCs were transferred to CNP-free IVM medium (containing 10 ng/mL epidermal growth factor [EGF]) to induce maturation. After incubation for 16 hr, COCs were denuded of cumulus cells by treatment with 0.03% hyaluronidase to obtain MII oocytes for future experiments.

## In vitro fertilization and embryo culture

Caudae epididymides from 12-week-old male C57BL-6J mice were lanced in a dish of in human tubal fluid (HTF) medium to release sperm, followed by being capacitated for 1 hr (37°C under an atmosphere of 5% $CO_2$ in air with maximum humidity). Matured oocytes were incubated with capacitated sperm at a concentration of $4 \times 10^5$/mL in 100 µL HTF for 6 hr at 37°C, 5% $CO_2$. The presence of two pronuclei was scored as successful fertilization. The embryos were cultured in KSOM under mineral oil at 37°C in 5% $CO_2$ and saturated humidity.

## Preantral follicle isolation and culture

Ovaries were removed after the animals had been killed by cervical dislocation and preantral follicles were mechanically isolated using 26-gauge needles. Then, the preantral follicles (80–100 µm diameter) that were enclosed by an intact basal membrane were collected, distributed randomly, and cultured individually in 96-well tissue culture plates for up to 6 d at 37°C in a humidified atmosphere of 5% $CO_2$ in air. The basic culture medium consisted of MEM-α supplemented with 1 mg/mL BSA, 1% ITS (5 µg/mL insulin, 5 µg/mL transferrin, 5 ng/mL selenium; Sigma), 100 µg/mL sodium pyruvate, and 1% penicillin/streptomycin sulfate (Sigma) in the absence (control) or presence of 100 nM CNP or 10 µM H89. The dose of CNP was selected based on the published literature (*Xi et al., 2019*). Half the medium was replaced with fresh medium and follicles were photographed every other day, and follicle diameter was measured using ImageJ at each time point.

## Immunofluorescent staining

Oocytes were fixed in 4% paraformaldehyde in PBS for 30 min at room temperature, permeabilized with 0.5% Triton X-100 for 20 min, then blocked with 1% BSA in PBS for 1 hr at room temperature. The oocytes were incubated with primary antibodies (Alexa Fluor 488 Conjugate anti-α-tubulin monoclonal antibody, 1:200, Cell Signaling, Cat#35652; rabbit anti-Tom20 antibody, 1:100, Cell Signaling, Cat#sc-42406; mouse anti-Parkin antibody, 1:100, Santa Cruz, Cat#sc-32282) at 4°C overnight, and then the oocytes were extensively washed with wash buffer (0.1% Tween 20 in PBS), probed with Alexa Fluor 488 goat anti-rabbit IgG (1:200, Thermo Fisher Scientific, A21206) or Alexa Fluor 594 donkey anti-mouse IgG (1:200, Abcam, ab150108) in a dark room for 1 hr at room temperature. Then oocytes were counterstained with DAPI (10 µg/mL) at room temperature for 10 min. Finally, samples were mounted on glass slides and viewed under the confocal microscope (Nikon A1R-si).

## MitoTracker, ER-Tracker, and Golgi-Tracker Red Staining

Oocytes were incubated with MitoTracker Red (1:2000, Beyotime Biotechnology, Shanghai, China), ER-Tracker Red (1:3000, Beyotime Biotechnology), and Golgi-Tracker Red (1:50, Beyotime Biotechnology) in M2 medium for 30 min at 37°C in a 5% $CO_2$ and saturated humidity. Then, the oocytes were

counterstained with DAPI (10 μg/mL) for 5 min at 37°C in a 5% $CO_2$ and saturated humidity, and finally, the samples were washed three times with M2 medium and examined with a confocal laser scanning microscope (Nikon A1R-si).

## Mitochondrial membrane potential (ΔΨm) measurement

Oocyte mitochondrial membrane potential was evaluated using Mito-Probe JC-1 Assay Kit (Beyotime Institute of Biotechnology, Shanghai, China). Briefly, oocytes were incubated with 2 μM JC-1 in M2 medium for 30 min at 37°C in a 5% $CO_2$ and saturated humidity, and finally, the samples were washed three times with M2 medium and examined with a confocal laser scanning microscope (Nikon A1R-si). JC-1 dye exhibits a fluorescence emission of green (529 nm) and red (590 nm). Thus, the red/green fluorescence intensity ratio was measured to indicate mitochondrial depolarization. Oocytes mitochondrial membrane potential (ΔΨm) measurements were performed as our previous report (*Zhang et al., 2020b*).

## Monitoring of ROS levels in oocytes

The amount of ROS in oocytes was processed with 10 μM oxidation-sensitive fluorescent probe DCFH (Beyotime Institute of Biotechnology) for 30 min at 37°C in M2 medium. Then oocytes were washed three times with M2 medium and placed on glass slides for image capture under a confocal microscope (Nikon A1R-si).

For the determination of mitochondrial ROS (MitoSOX) generation by MitoSOX staining, GV oocytes were incubated in M2 media containing 5 μM MitoSOX Red (Thermo Fisher, M36008, Waltham) in humidified atmosphere for 10 min at 37°C. After washing three times in M2 media, oocytes were imaged under a confocal microscope (Nikon A1R-si).

## Measurement of the GSH/GSSG ratio

The GSH/GSSG ratio was measured with a GSSG/GSH Assay Kit (Beyotime Institute of Biotechnology) according to the manufacturer's instructions. Briefly, oocytes were lysed in 40 μL deproteinized buffer on ice for 10 min. The lysate was centrifuged at 12,000 × *g* for 5 min at 4°C. For GSSG measurement, the samples were incubated with GSH scavenge buffer for 60 min at 25°C to decompose GSH. Then, the samples were transferred to the 96-well plates and the absorbance was measured with a multimode plate reader (BioTek Epoch) at 412 nm.

## Measurement of the oocyte NADPH content

The oocyte NADPH contents were measured using a NADPH assay kit (Beyotime Institute of Biotechnology) according to the manufacturer's instructions. Briefly, 50–60 oocytes per group were lysed in 100 μL NADPH extraction buffer on ice for 20 min. After the samples were centrifuged at 12,000 × g for 5 min at 4°C, the supernatants were transferred to the 96-well plates (50 μL per well), and the absorbance was measured using a multimode plate reader (BioTek Epoch) at 450 nm. The amount of NADPH was determined using a calibration curve.

## Western blot analysis

Approximately 200 oocytes were lysed in RIPA buffer (solarbio, Beijing, China) supplemented with 1 mM protease inhibitor phenylmethylsulfonyl fluoride (PMSF, solarbio) on ice for 30 min. Samples were boiled at 100°C in a metal bath for 10 min in protein loading buffer (CoWin Biosciences, Beijing, China) and equal amount of proteins were separated by 10% SDS-PAGE gel and transferred to polyvinylidene fluoride (PVDF) membranes (Millipore, Bedford, USA). After transfer, the membranes were blocked in TBST that contained 3% BSA for 1 hr at room temperature, followed by incubation with primary antibodies at 4°C overnight (the primary antibodies were rabbit anti-GAPDH antibody, 1:2000, Cell Signaling, Cat#5174; rabbit anti-p62 antibody, 1:1000, Cell Signaling, Cat#23214; rabbit anti-Tom20 antibody, 1:1000, Cell Signaling, Cat#sc-42406; rabbit anti-LC3A/B antibody, 1:1000, Abcam, Cat#ab128025; rabbit anti-PINK1 antibody, 1:1000, Cell Signaling, Cat#6946; mouse anti-Parkin antibody, 1:1000, Santa Cruz, Cat#sc-32282). The secondary antibodies were incubated for 1 hr at room temperature, then the membrane signals were visualized by a chemiluminescent HRP substrate reagent (Bio-Rad Laboratories, Hercules, CA), and images were captured with Tanon5200

Imaging System (Biotanon, Shanghai, China). The band intensity was assessed with ImageJ software and normalized to that of GAPDH.

### Transmission electron microscope (TEM)

Oocytes were prefixed with 3% glutaraldehyde, refixed with 1% osmium tetroxide, dehydrated in acetone, and embedded in Ep812 (Can EM Ltd.). Semithin sections were stained with toluidine blue for optical positioning, and ultrathin sections were made with a diamond knife and observed by a JEM-1400FLASH transmission electron microscope (JEOL) after staining with uranyl acetate and lead citrate.

### Evaluation of total ATP content

The ATP content of oocytes was detected with an ATP Bioluminescence Assay Kit (Beyotime Institute of Biotechnology). The oocytes were lysed with 50 µL of ATP lysis buffer on ice, centrifuged at 12,000 × $g$ at 4°C for 5 min, and the supernatants were transferred to a 96-well black culture plate. Then, the samples and standards were read with a Multimode Microplate Reader (Tecan Life Sciences). Finally, the ATP level was calculated according to the standard curve.

### RNA sequencing and analysis

GV-stage oocytes were collected from three young, three aged, and three CNP-treated aged mice. mRNA samples were collected from five oocytes from the same mouse of each group. The mRNA was directly reverse-transcribed through oligo dT. The reverse-transcribed cDNA was amplified, and the cDNA was cut by Tn5 transposase digestion, and linkers were added to obtain the required sequencing library. The constructed library was entered into the sequencing program after passing through an Agilent 2100 Bioanalyzer and RT-PCR quality control. The PE100 sequencing strategy was used to assess gene expression changes at the transcription level. Dr. Tom (Dr. Tom is a web-based solution that offers convenient analysis, visualization, and interpretation of various types of RNA data, https://www.bgi.com/global/service/dr-tom) was used for difference analysis, GO analysis, KEGG analysis, and other analyses.

### Reverse transcriptase quantitative PCR (RT-qPCR) analysis

Total RNA from oocytes was extracted using MiniBEST Universal RNA Extraction Kit (TaKaRa, Dalian, China) and reverse-transcribed to synthesize cDNA using a PrimeScript RT Master Mix reverse transcription kit (TaKaRa) according to the manufacturer's instructions. RT-qPCR quantitation of mRNAs was performed using TB Green Premix Ex Taq II (TaKaRa) with Applied Biosystems StepOnePlus Real-Time PCR System (Thermo Fisher Scientific, MA) using the following parameters: 95°C for 1 min, followed by 40 cycles at 95°C for 5 s and 60°C for 34 s. The PCR primers used in this study are shown in *Supplementary file 1* primers sequences table. Transcript levels were normalized to those of the housekeeping gene *Gapdh*. The CT value was used to calculate the fold change using the $2^{-\triangle\triangle Ct}$ method. Each experiment was repeated independently at least thrice.

### Statistical analysis

Statistical analyses were performed using GraphPad Prism 8.00 software (GraphPad, CA). Differences between two groups were assessed using the *t*-test. Data from at least three biological repeats are reported as means ± SEM. Results of statistically significant differences are denoted by asterisk: *$p < 0.05$ **$p < 0.01$, ***$p < 0.001$, and ****$p < 0.0001$.

## Acknowledgements

We thank other members of the Ma laboratory for their helpful discussion on the manuscript. This work was supported in part by the National Natural Science Foundation of China (31772818) and Postdoctoral Science Foundation of Shaanxi Province of China (2023BSHYDZZ89).

## Additional information

### Funding

| Funder | Grant reference number | Author |
|---|---|---|
| National Natural Science Foundation of China | 31772818 | Baohua Ma |
| Postdoctoral Science Foundation of Shaanxi Province of China | 2023BSHYDZZ89 | Hui Zhang |

The funders had no role in study design, data collection and interpretation, or the decision to submit the work for publication.

### Author contributions

Hui Zhang, Resources, Data curation, Formal analysis, Funding acquisition, Validation, Investigation, Visualization, Methodology, Writing – original draft, Project administration, Writing – review and editing; Chan Li, Data curation, Formal analysis, Supervision, Validation, Investigation, Visualization, Methodology, Writing – original draft, Writing – review and editing; Qingyang Liu, Data curation, Formal analysis, Visualization, Methodology, Writing – review and editing; Jingmei Li, Validation, Investigation, Visualization, Methodology, Writing – review and editing; Hao Wu, Data curation, Methodology, Writing – review and editing; Rui Xu, Ming Cheng, Formal analysis, Methodology; Yidan Sun, Conceptualization, Writing – review and editing; Xiaoe Zhao, Formal analysis, Writing – review and editing; Menghao Pan, Formal analysis, Supervision; Qiang Wei, Supervision, Validation, Visualization, Project administration, Writing – review and editing; Baohua Ma, Supervision, Funding acquisition, Validation, Visualization, Project administration, Writing – review and editing

### Author ORCIDs

Jingmei Li ⓘ http://orcid.org/0009-0009-9423-8696
Baohua Ma ⓘ http://orcid.org/0000-0002-8332-4275

### Ethics

The experimental protocols and mice handling procedures were reviewed and approved by the Institutional Animal Care and Use Committee of the College of Veterinary Medicine, Northwest A&F University (No. 2018011212).

Reviewer #1 (Public Review): https://doi.org/10.7554/eLife.88523.3.sa1
Reviewer #2 (Public Review): https://doi.org/10.7554/eLife.88523.3.sa2
Author Response https://doi.org/10.7554/eLife.88523.3.sa3

## Additional files

### Supplementary files

- Supplementary file 1. Primers sequences table.
- MDAR checklist

### Data availability

All data generated or analysed during this study are included in the manuscript and supporting files. The raw RNAseq data have been provided as *Figure 5—source data 1*.

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
