## [Editor Report · eLife assessment]

This study presents **valuable** findings on the impact of C-natriuretic peptide (CNP) treatment in vivo on the fertility of aged mice. **Solid** data indicate CNP induces the cAMP-PKA pathway, causing reduced recruitment of Parkin protein to mitochondria in oocytes, resulting in reduced mitophagy, which may be significant for increased mitochondrial bioenergetics and improved cytoplasmic and nuclear maturation. The authors make additional claims regarding the mechanisms by which CNP impacts oocyte quality in vivo for which the evidence is inconclusive. This work will be of interest to reproductive biologists and clinical infertility specialists.

---

## [Referee Report · Reviewer #1 (Public Review)]

It has been shown previously that maternal aging in mice is associated with an increase in accumulation of damaged mitochondria and activation of parkin-mediated autophagy (see DOI: 10.1080/15548627.2021.1946739). It has also been shown that C-natriuretic peptide (CNP) regulates oocyte meiotic arrest and that its use during in vitro oocyte maturation can improve parameters associated with decreased oocyte quality. Here the authors tested whether use of CNP treatment in vivo could improve oocyte quality and fertility of aged mice, for which they provided convincing evidence. They also attempted to determine how CNP improves oocyte developmental competence. They showed a correlation between CNP use in vivo and the appearance (and some functional qualities) of cytoplasmic organelles more closely approximating those of oocytes from young mice. However, this correlation could not be interpreted to imply causation. Additional experiments performed using CNP during in vitro maturation were not properly controlled and so are not possible to interpret.

A strength of the manuscript is that the authors use an in vivo treatment to improve oocyte quality rather than just using CNP during oocyte maturation in vitro as has been done previously. This strategy provides more potential for improving oocyte quality - over the course of oocyte growth and maturation - rather than just the final few hours of maturation alone. This strategy also has the potential to be translated into a more generally useful clinical therapeutic method that using CNP during in vitro maturation. However, it is difficult to glean information regarding how CNP might have its effects in vivo. A range of models are used in the manuscript with a mix of in vivo studies with in vitro experiments, which results in some disconnect between systemic CNP and its reported intrafollicular action as well as in the short-term versus longer-term actions of CNP on oocyte quality. Specifically, CNP was shown to be reduced in the plasma of aged mice, but this was not shown in the granulosa cells, which are the reported source of CNP that acts on oocytes. Whether the ovarian source of CNP is reduced in aged females was not demonstrated, and CNP is not known to act on oocytes through an endocrine effect. In vivo treatments with CNP by i.p. injection were performed, but the dose (120 ug/kg) and time (14 days) of treatment were not validated by any prior experiments to give them physiological relevance.

Weaknesses:

1. The Results section is not always clear regarding what CNP treatment was done - in vivo injections or in vitro maturation. For example, what is the difference, if any, between Figures 2C-D and Figures S2A-B?

This remains unclear in the revised manuscript.

2. Immature oocytes from aged females (~1 year) were treated with a two-step culture system with a pre-IVM step with CNP. Controls included oocytes from young (6-8 weeks) females or oocytes from aged females treated by conventional IVM. The description of these methods suggests that control oocytes did not receive an equivalent pre-IVM culture, hence the relevance of comparisons of CNP-treated versus control oocyte is questionable. This concern has not been addressed in the revised manuscript. It was observed that aged oocytes pre-cultured in CNP improved polar body extrusion rates and meiotic spindle morphology compared to oocytes in conventional IVM, as has been well established. The description of statistical methods does not make clear whether the PBE rate in CNP-treated old oocytes remained significantly lower than young controls.

This concern has not been addressed in the revised manuscript.

3. The main effect of the CNP 2-week treatment appears to be increasing the number of follicles that grow into secondary and antral stages, but there is no attempt made to discover the mechanism by which this occurs and therefore to understand why there might be an increase in the number of ovulated eggs, quality of the eggs, and litter size. It is also not clear how an intraperitoneal injection can guarantee its effectiveness because the half-life of CNP is very short, only a few minutes.

This concern has not been addressed in the revised manuscript.

4. Meiotic spindle morphology, as well as a number of putative markers of cytoplasmic maturation are also suggested to be improved after pre-culture with CNP. In each case a subjective interpretation of "normal" morphology of these markers is derived from observations of the young controls and the proportions of oocytes with normal or abnormal appearance is evaluated. However, parameters that define abnormal patterns of these markers appear to be subjective judgements, and whether these morphological patterns can be mechanistically attributed to the differences in developmental potential cannot be concluded.

This concern has not been addressed in the revised manuscript.

5. In addition to the localization patterns of mitochondria, the mitochondrial membrane potential, oocyte ATP content and ROS levels were assessed through more objective quantitative methods. These are well known to be defective in oocytes of aged females and CNP treatment improved these measures. Mitochondrial dysfunction is the most obvious link between oocyte apoptosis, autophagy, cytoplasmic organelle miss-localization and aberrant spindle morphology. Among the most intriguing results is the finding that CNP mediated a cAMP-dependent protein kinase (PKA) dependent reduction in mitochondrial autophagy mediators PINK and Parkin and reduced the recruitment of Parkin to mitochondria in oocytes. However, it may not be possible to directly link this observation to the improvements in IVM oocyte quality, since PINK/Parkin assessments were performed in oocytes from cultured follicles treated with CNP for 6 days.

This weakness has not been addressed in the revised manuscript.

6. The gold standard assay for oocyte quality is embryo transfer and live birth. The authors assessed the impact of maturing oocytes in vitro in the presence of CNP on oocyte quality by less robust assays (e.g., preimplantation embryo development in vitro), so the impact on oocyte quality is less certain.

This weakness has not been addressed in the revised manuscript.

7. The numbers of embryos should have been corrected for the number of eggs fertilized as a starting point so that the percentage that developed to each stage could be expressed as a percentage of successfully fertilized eggs rather than overall percentages. As currently shown in the Figures and described in the Legend, there is no information regarding what the percentage on the y-axis means. For example, does Figure 4B show the number of 2C embryos divided by the number of eggs inseminated? Or is it divided by the number of successfully fertilized eggs, and if so, how was that assessed?

There is no additional information provided in the revised manuscript to address these concerns.

8. When fewer eggs are fertilized, the numbers of embryos per group are lower and so the impact of culturing multiple embryos together is lost. As a result, it is possible that culture conditions rather than oocyte quality drove the differences in the numbers of embryos that achieved each stage of development.

This concern has not been addressed in the revised manuscript. Similar numbers of oocytes were cultured together, but not similar numbers of fertilized oocytes, or embryos.

9. Not all claims in the Discussion are supported by the evidence provided. For example, "In addition, the findings demonstrated that CNP improved cytoplasmic maturation events by maintaining normal CG, ER and Golgi apparatus distribution and function in aged oocytes" but it was never demonstrated that the altered distribution had any functional impact.

This concern has not been addressed in the revised manuscript.

10. Incompleteness and errors in the Methods section reduce confidence in many of the results reported.

This concern has not been addressed in the revised manuscript.

11. The methods used for Statistical Analysis are never explained in either the Methods or the Figure legends. It is unclear whether appropriate analyses were done, and it is frequently unclear what was the sample size and how many times a particular experiment was repeated. These weaknesses detract from confidence in the data.

This concern has not been addressed adequately in the revised manuscript.

---

## [Referee Report · Reviewer #2 (Public Review)]

The authors found that the age-related reduction in the serum CNP concentration was highly correlated with decreased oocyte quality. Treatment with exogenous CNP promoted follicle growth and ovulation in aged mice and enhanced meiotic competency and fertilization ability. The cytoplasmic maturation of aged oocytes was thoroughly improved by CNP treatment. CNP treatment also ameliorated DNA damage and apoptosis caused by ROS accumulation in aged oocytes. CNP reversed the defective phenotypes in aged oocytes by alleviating oxidative damage and suppressing excessive PINK1/Parkin-mediated mitophagy. CNP functioned as a cAMP/PKA pathway modulator to decrease PINK1 stability and inhibit Parkin recruitment. CNP may be used to improve the overall success rates of clinically assisted reproduction in older women.

The author has modified the text and the level of the article has been improved. Additional experiments will further enhance the credibility of the article.

1）The control also needs to be pre-cultured as that in CNP treatment.

2）The mechanism is done 6 days later after CNP treatment. It is hard to know whether it is direct or indirect.

---

## [Author Response]

The following is the authors’ response to the original reviews.

**Combined Public Review:**
It has been shown previously that maternal aging in mice is associated with an increase in accumulation of damaged mitochondria and activation of parkin-mediated autophagy (see DOI: 10.1080/15548627.2021.1946739). It has also been shown that C-natriuretic peptide (CNP) regulates oocyte meiotic arrest and that its use during in vitro oocyte maturation can improve parameters associated with decreased oocyte quality. Here the authors tested whether use of CNP treatment in vivo could improve oocyte quality and fertility of aged mice, for which they provided convincing evidence. They also attempted to determine how CNP improves oocyte developmental competence. They showed a correlation between CNP use in vivo and the appearance (and some functional qualities) of cytoplasmic organelles more closely approximating those of oocytes from young mice. However, this correlation could not be interpreted to imply causation. Additional experiments performed using CNP during in vitro maturation were not properly controlled and so are not possible to interpret.A strength of the manuscript is that the authors use an in vivo treatment to improve oocyte quality rather than just using CNP during oocyte maturation in vitro as has been done previously. This strategy provides more potential for improving oocyte quality - over the course of oocyte growth and maturation - rather than just the final few hours of maturation alone. This strategy also has the potential to be translated into a more generally useful clinical therapeutic method that using CNP during in vitro maturation. However, it is difficult to glean information regarding how CNP might have its effects in vivo. A range of models are used in the manuscript with a mix of in vivo studies with in vitro experiments, which results in some disconnect between systemic CNP and its reported intrafollicular action as well as in the short-term versus longer-term actions of CNP on oocyte quality. Specifically, CNP was shown to be reduced in the plasma of aged mice, but this was not shown in the granulosa cells, which are the reported source of CNP that acts on oocytes. Whether the ovarian source of CNP is reduced in aged females was not demonstrated, and CNP is not known to act on oocytes through an endocrine effect. In vivo treatments with CNP by i.p. injection were performed, but the dose (120 ug/kg) and time (14 days) of treatment were not validated by any prior experiments to give them physiological relevance.Thank you for the summary and for highlighting our manuscript’s strengths and weaknesses.Weaknesses:1. There are errors in the manuscript writing that make the Results difficult to follow. Reference to the Figures in the Results section does not match what is shown in the Figure panels. For example, the Results text reports differences in CNP levels in aged and young mice shown in Figure 1C, but the relevant panel is actually shown in Figure 1F. Other Figures have the same problem.

Thanks for the valuable suggestion. All the mistakes have been corrected in the revised manuscript.

1. The Results section is not always clear regarding what CNP treatment was done - in vivo injections or in vitro maturation. For example, what is the difference, if any, between Figures 2C-D and Figures S2A-B?

Thank you for pointing out the potential confusion regarding the experimental procedures in Figures 2C-D and Figures S2A-B. In the revised manuscript, we have included additional explanations to clarify that Figures 2C-D represent in vivo injections, while Figures S2A-B depict in vitro maturation. In brief, the results presented in the Supplementary Material (Figures S1-S7) are derived from in vitro CNP treatment.

1. Immature oocytes from aged females (~1 year) were treated with a two-step culture system with a pre-IVM step with CNP. Controls included oocytes from young (6-8 weeks) females or oocytes from aged females treated by conventional IVM. The description of these methods suggests that control oocytes did not receive an equivalent pre-IVM culture, hence the relevance of comparisons of CNP-treated versus control oocyte is questionable. It was observed that aged oocytes pre-cultured in CNP improved polar body extrusion rates and meiotic spindle morphology compared to oocytes in conventional IVM, as has been well established. The description of statistical methods does not make clear whether the PBE rate in CNP-treated old oocytes remained significantly lower than young controls.

Statistical analyses were performed using GraphPad Prism 8.00 software (GraphPad, CA, United States). Differences between two groups were assessed using the t-test. Indeed, CNP is unlikely to fully restore the PB1 rate in aged mice to the same level as in the young group. PB1 rate in CNP-treated aged oocytes remained significantly lower than young controls (P<0.05).

1. The main effect of the CNP 2-week treatment appears to be increasing the number of follicles that grow into secondary and antral stages, but there is no attempt made to discover the mechanism by which this occurs and therefore to understand why there might be an increase in the number of ovulated eggs, quality of the eggs, and litter size. It is also not clear how an intraperitoneal injection can guarantee its effectiveness because the half-life of CNP is very short, only a few minutes.

The 2-week treatment of CNP had a significant impact, leading to an increase in the number of follicles progressing to secondary and antral stages, as well as an increase in the number of ovulated eggs, improved egg quality, and enhanced litter size. Previous studies (references: 10.1530/REP-18-0470; 10.1210/me.2012-1027) have demonstrated the crucial role of CNP as an upstream regulator in stimulating preantral follicle growth and promoting the ovulation rate. These studies have also identified the influence of CNP on the expression of key ovarian genes involved in cell growth and steroidogenic enzymes. Consistent with these findings, our study provides further evidence supporting CNP as a critical regulator of preantral follicle growth and oocyte quality. Furthermore, it is important to note that oocyte-derived paracrine factors play essential roles in follicular development. CNP may regulate the communication between oocytes and somatic cells, contributing to folliculogenesis and follicular development. We are considering this aspect for further investigation in another ongoing study.

To ensure the effectiveness of CNP, given its short half-life (a few minutes), aged mice (58 weeks old) received daily intraperitoneal injections of CNP (120 μg/kg body weight; Cat#B5441, ApexBio) for a duration of 14 days.

1. Meiotic spindle morphology, as well as a number of putative markers of cytoplasmic maturation are also suggested to be improved after pre-culture with CNP. In each case a subjective interpretation of "normal" morphology of these markers is derived from observations of the young controls and the proportions of oocytes with normal or abnormal appearance is evaluated. However, parameters that define abnormal patterns of these markers appear to be subjective judgements, and whether these morphological patterns can be mechanistically attributed to the differences in developmental potential cannot be concluded.

Oocyte cytoplasmic maturation involves a remarkable reorganization of the oocyte cytoplasm, encompassing the movement of vesicles, mitochondria, Golgi apparatus, and endoplasmic reticulum. This dynamic process occurs during the transitions from the germinal vesicle breakdown (GVBD) stage to the metaphase I (MI), polar body extrusion (PBE), and metaphase II (MII) stages (reference: 10.1093/humupd/dmx040). In our study, we observed that CNP treatment partially rescued cytoplasmic maturation events in aged oocytes by maintaining normal distribution patterns of cortical granules (CG), endoplasmic reticulum (ER), and Golgi apparatus. However, further experiments are needed to investigate the specific action of CNP on the function of CG, ER, and Golgi apparatus. These experiments are beyond the scope of this manuscript, but we acknowledge the importance of this aspect and will consider it for future research. In this study, our main focus was to examine the effects of CNP on mitochondria distribution and function. Therefore, we analyzed the localization patterns of mitochondria, mitochondrial membrane potential, oocyte ATP content, and ROS levels. These experiments were aimed at elucidating the impact of CNP on mitochondrial dynamics and metabolism, which are crucial for oocyte quality and development.

1. In addition to the localization patterns of mitochondria, the mitochondrial membrane potential, oocyte ATP content and ROS levels were assessed through more objective quantitative methods. These are well known to be defective in oocytes of aged females and CNP treatment improved these measures. Mitochondrial dysfunction is the most obvious link between oocyte apoptosis, autophagy, cytoplasmic organelle miss-localization and aberrant spindle morphology. Among the most intriguing results is the finding that CNP mediated a cAMP-dependent protein kinase (PKA) dependent reduction in mitochondrial autophagy mediators PINK and Parkin and reduced the recruitment of Parkin to mitochondria in oocytes. However, it may not be possible to directly link this observation to the improvements in IVM oocyte quality, since PINK/Parkin assessments were performed in oocytes from cultured follicles treated with CNP for 6 days.

The beneficial effects of CNP on oocyte quality have been extensively demonstrated through in vivo experiments (Figure 1 and 4) and “two-step” in vitro culture experiments (Figure S1 and S7). In this study, our primary focus is to analyze the signaling pathway and mechanism by which CNP inhibits mitophagy in oocytes. Previous studies have highlighted the significant role of cAMP-PKA activity in reducing mitochondrial recruitment of Parkin and mitophagy (reference: 10.1038/s42003-020-01311-7). Consistent with these findings, our study revealed that aged oocytes exhibited lower concentrations of cAMP compared to young oocytes. However, upon administration of CNP, we observed a substantial increase in intraoocyte cAMP levels. To investigate the involvement of PKA in CNP-mediated oocyte mitophagy, we conducted further experiments. We isolated preantral follicles (80-100 µm diameter) from the ovaries of aged mice and subjected them to in vitro culture with either 100 nM CNP or a combination of 100 nM CNP and 10 µM H89, a PKA inhibitor. Monitoring the growth dynamics of the follicles revealed that treatment with 100 nM CNP significantly increased follicle diameter, while H89 treatment inhibited the promotive effect of CNP on preantral follicle growth (Figure 6 K and L). Western blot analysis demonstrated that CNP supplementation led to a significant decrease in PINK1 and Parkin expression levels, which were abrogated by H89 treatment (Figure 6 M-O). It is well-established that the cAMP-PKA pathway plays a crucial role in inhibiting Parkin recruitment to damaged mitochondria (Akabane et al., 2016). Therefore, we aimed to investigate whether PKA inhibition regulates Parkin recruitment. To assess the effects of CNP on mitochondria, we performed double staining for Parkin and translocase of outer mitochondrial membrane 20 (TOMM20). The results clearly demonstrated that CNP inhibited the mitochondrial localization of Parkin, while PKA inhibition with H89 led to Parkin translocation to mitochondria, as indicated by the overlap of the two staining signals (Figure 6 P and Q). Collectively, our data suggest that the suppression of Parkin recruitment through the cAMP-PKA axis represents an important mechanism underlying the protective effect of CNP against oxidative injury in maternally aged mouse oocytes.

1. The gold standard assay for oocyte quality is embryo transfer and live birth. The authors assessed the impact of maturing oocytes in vitro in the presence of CNP on oocyte quality by less robust assays (e.g., preimplantation embryo development in vitro), so the impact on oocyte quality is less certain.

We appreciate the Revierer’s suggestion to assay live birth rates by transfer embryos obtained from IVM oocytes. However, we decided not to pursue this option for this revision because of the current technical challenges that make it difficult to get a precise result of live birth rates from IVM oocyte. Thank you for your very valuable suggestion, we have discovered the shortcomings in my current work, and I will follow your suggestions in my future work to improve the level of scientific research and achieve more results.

1. The terminology used to describe many of the Results exaggerates the findings. For example, the authors claim that many of their immunofluorescent markers of the various organelles have a pattern that is "restored" by CNP. However, in most cases the pattern is "improved" toward the control condition but is not fully restored.

We acknowledge the confusion caused by the wording of the mechanism of action of CNP in the original version. In the resubmission, we have made significant improvements by providing critical information that clarifies the action of CNP. We believe that these revisions will enhance the understanding of the mechanism of CNP and its implications. Thank you for pointing out this issue, and we appreciate your feedback in helping us improve the clarity of our work.

1. The numbers of embryos should have been corrected for the number of eggs fertilized as a starting point so that the percentage that developed to each stage could be expressed as a percentage of successfully fertilized eggs rather than overall percentages. As currently shown in the Figures and described in the Legend, there is no information regarding what the percentage on the y-axis means. For example, does Figure 4B show the number of 2C embryos divided by the number of eggs inseminated? Or is it divided by the number of successfully fertilized eggs, and if so, how was that assessed?

The embryonic development rates (Figure 4 B-F) were calculated based on the total number of oocytes, and the percentages of oocytes that developed to each stage were expressed as overall percentages.

1. When fewer eggs are fertilized, the numbers of embryos per group are lower and so the impact of culturing multiple embryos together is lost. As a result, it is possible that culture conditions rather than oocyte quality drove the differences in the numbers of embryos that achieved each stage of development.

The embryonic development rate was calculated based on the total number of oocytes. Each group included a minimum of 50 oocytes with three replicates (Young: 51, aged: 53, CNP+aged: 50). The embryo culture conditions were consistent across all groups.

1. Not all claims in the Discussion are supported by the evidence provided. For example, "In addition, the findings demonstrated that CNP improved cytoplasmic maturation events by maintaining normal CG, ER and Golgi apparatus distribution and function in aged oocytes" but it was never demonstrated that the altered distribution had any functional impact.

Oocyte cytoplasmic maturation involves a remarkable reorganization of the oocyte cytoplasm, including the movement of vesicles, mitochondria, Golgi apparatus, and endoplasmic reticulum. Extensive remodeling and repositioning of intracellular organelles occur during the transitions from GVBD to MI, PBE, and MII stages (10.1093/humupd/dmx040). Our findings indicate that CNP partially rescued cytoplasmic maturation events in aged oocytes by preserving normal distribution of CG, ER, and Golgi apparatus, as well as maintaining mitochondrial function. We acknowledge the importance of considering the impact of CNP on the function of CG, ER, and Golgi apparatus for future research. In summary, these findings demonstrate that CNP improves cytoplasmic maturation events in aged oocytes by facilitating the reorganization of CG, ER, and Golgi apparatus.

1. Incompleteness and errors in the Methods section reduce confidence in many of the results reported.

We will enhance the readability of the entire Methods section for the resubmission.

1. The methods used for Statistical Analysis are never explained in either the Methods or the Figure legends. It is unclear whether appropriate analyses were done, and it is frequently unclear what was the sample size and how many times a particular experiment was repeated. These weaknesses detract from confidence in the data.

Statistical analyses were performed using GraphPad Prism 8.00 software (GraphPad, CA, United States). Differences between two groups were assessed using the t-test. Data were reported as means ± SEM. Results of statistically significant differences were denoted by asterisk. (P < 0.05 denoted by *, P < 0.01 denoted by **, P < 0.001 denoted by ***, and P < 0.0001 denoted by ****).

Recommendations for the authors: please note that you control which revisions to undertake from the public reviews and recommendations for the authors1. The introduction does not provide critical information regarding what is already known about the mechanism of action of CNP, what other tissues are impacted by CNP treatment, and how it might affect oocyte growth. Providing this information would make it much easier to understand what is novel about the current manuscript.

We acknowledge that the mechanism of action of CNP was unclear in the original version. We have now included essential information to clarify the action of CNP.

1. Comparison of the RNAseq dataset to robust datasets from young vs aged mice would strengthen the analysis (e.g., the dataset in DOI: 10.1111/acel.13482).

Thank you for your professional suggestion. According to the suggestion from you, we will make comparison of the RNAseq dataset to robust datasets from young vs aged mice in my future work.

1. Please explain what is "Dr. Tom" that was used for RNA sequencing analysis, in the Methods.

Dr. Tom is a web-based solution that offers convenient analysis, visualization, and interpretation of various types of RNA data, including mRNA, miRNA, and lncRNA. It also supports the interpretation of single-cell RNA-seq data and WGBS data. Developed by a team of expert scientists and bioinformaticians at BGI, who have extensive experience in numerous research projects, Dr. Tom provides a wide range of intuitive and interactive data visualization tools tailored to save time in conducting differential expression or pathway analysis research. Moreover, its powerful analysis tools and advanced algorithms enable users to extract new insights and derive additional value from their data beyond what is available through standard RNA analysis services. The integration of data from leading databases worldwide allows users to reference and cross-check their results and findings. Dr. Tom is already trusted by tens of thousands of scientists and researchers, serving as a valuable and essential tool alongside their own internal data curation and analysis efforts. To learn more, please visit: Dr. Tom website https://www.bgi.com/global/service/dr-tom.

1. The Results state that single-cell transcriptomics was performed, but the Methods state that 5 oocytes were collected from each mouse. The actual Method used should be clarified.

Single-cell RNA-seq is a powerful technique that enables digital transcriptome analysis at the single-cell level using deep-sequencing methods. With this approach, even a single cell can be isolated and processed through various steps to generate sequencing libraries. Given the limited availability of oocyte samples, we employed a single-cell RNA-seq library construction protocol, allowing us to analyze the transcriptomes of individual oocytes. As a result, we collected and analyzed five oocytes from each mouse in our study.

1. The raw RNAseq data should be deposited into a publicly accessible database and reported by an accession number. It is not sufficient to state that the data is included in the manuscript and supporting information.

The RNA-seq data has been submitted as supporting information and is now accessible to all readers.

1. The image in Figure 1G is not very clear.

Thank you for bringing this to our attention. We will enhance the readability of all our figures for the resubmission.